# DEAD-Box RNA Helicases DDX3X and DDX5 as Oncogenes or Oncosuppressors: A Network Perspective

**DOI:** 10.3390/cancers14153820

**Published:** 2022-08-06

**Authors:** Massimiliano Secchi, Camilla Lodola, Anna Garbelli, Silvia Bione, Giovanni Maga

**Affiliations:** Institute of Molecular Genetics, IGM CNR “Luigi Luca Cavalli-Sforza”, Via Abbiategrasso 207, 27100 Pavia, Italy

**Keywords:** DEAD-box RNA helicases, cancer, DDX3X, DDX5, oncogene, oncosuppressor

## Abstract

**Simple Summary:**

The transformation of a normal cell into a cancerous one is caused by the deregulation of different metabolic pathways, involving a complex network of protein–protein interactions. The cellular enzymes DDX3X and DDX5 play important roles in the maintenance of normal cell metabolism, but their deregulation can accelerate tumor transformation. Both DDX3X and DDX5 interact with hundreds of different cellular proteins, and depending on the specific pathways in which they are involved, both proteins can either act as suppressors of cancer or as oncogenes. In this review, we summarize the current knowledge about the roles of DDX3X and DDX5 in different tumors. In addition, we present a list of interacting proteins and discuss the possible contribution of some of these protein–protein interactions in determining the roles of DDX3X and DDX5 in the process of cancer proliferation, also suggesting novel hypotheses for future studies.

**Abstract:**

RNA helicases of the DEAD-box family are involved in several metabolic pathways, from transcription and translation to cell proliferation, innate immunity and stress response. Given their multiple roles, it is not surprising that their deregulation or mutation is linked to different pathological conditions, including cancer. However, while in some cases the loss of function of a given DEAD-box helicase promotes tumor transformation, indicating an oncosuppressive role, in other contexts the overexpression of the same enzyme favors cancer progression, thus acting as a typical oncogene. The roles of two well-characterized members of this family, DDX3X and DDX5, as both oncogenes and oncosuppressors have been documented in several cancer types. Understanding the interplay of the different cellular contexts, as defined by the molecular interaction networks of DDX3X and DDX5 in different tumors, with the cancer-specific roles played by these proteins could help to explain their apparently conflicting roles as cancer drivers or suppressors.

## 1. Introduction

The DEAD-box RNA helicase family is a class of enzymes that belongs to the helicase superfamily 2 [1]. Due to the presence of a characteristic Asp-Glu-Ala-Asp (DEAD) motif, DEAD-box proteins have been identified in almost all organisms ranging from bacteria to humans. These highly conserved enzymes use the energy of ATP hydrolysis to unwind the double-stranded RNA (dsRNA) molecules [2]. All the proteins of the family share two RecA-like domains that contain a conserved core of structural motifs that mediate ATP binding, ATP hydrolysis, RNA binding, and RNA unwinding (Figure 1). A flexible linker connects the two RecA-like domains, favoring their orientation changes that are necessary for the transition between the open and the close conformation states of the enzymes. This is critical for the function of the helicases as the cooperative binding of dsRNA and ATP leads to the adoption of the close conformation in which the RecA-like domains are in close proximity, allowing the opening of dsRNA.

Then, ATP hydrolysis triggers the release of the displaced RNA strand, restoring the original open conformation with the two RecA-like domains separated [6]. In contrast to the conserved helicase core, the N- and C-terminal extensions are variable and divergent in length and composition. These domains control the interaction of the RNA helicases with particular RNA targets or with other proteins and are thought to confer functional specificity to these enzymes [7]. As a result of their diversity of binding and function, DEAD-box RNA helicases are involved in almost all aspects of RNA metabolism, from transcription, mRNA processing and export to mRNA translation and miRNA biogenesis [8]. Moreover, in addition to duplex unwinding and the correlated ATPase activities, DEAD-box helicases can also function as assembly platforms for larger ribonucleoprotein complexes by binding to other proteins through the auxiliary domains or the helicase core. As a result of these interactions, these RNA helicases are involved in many biological processes, such as cell cycle regulation, stress response, apoptosis, innate immunity and virus infection, so they are considered to be more than simple RNA duplex unwinders [9]. Mutations in helicase genes have been linked to vital functions as they lead to different chromosomal instabilities associated with neurological and neurodevelopment disorders or to hereditary diseases. Furthermore, recent reports have highlighted that RNA helicases play a central role in the maintenance of genome stability by regulating the expression of genes that are important for DNA damage repair or by direct involvement in the processing of complex nucleic acid species such as DNA–RNA hybrids, R-loops and telomeric sequences [10]. Therefore, it is not surprising that DEAD-box helicases have been implicated in cellular proliferation and/or neoplastic transformation of various types of tumors and are considered a new attractive target for the development of novel pharmaceutical anticancer drugs [11]. Interestingly, the role of these enzymes in cancer development is rather controversial as they have been suggested to have tumor suppressor properties in some cancers or to act as oncogenes in others [12]. Here, we focus on two of the best-characterized members of the DEAD-box RNA helicase family, DDX3X and DDX5, for which extensive literature and proteomics data are available, summarizing their pro- and anti-proliferative roles in different malignancies. The two proteins show only 60% homology and several studies have suggested the possibility of their direct interaction. For example, they are both involved in miRNA metabolism, mRNA export, RNA splicing, viral replication and calmodulin-associated protein complexes. Recently, protein–protein interaction between these two DEAD-box RNA helicases has been demonstrated in the export of ribonucleoprotein complexes into the cytoplasm where DDX3X is required for the shuttling of DDX5 to the nucleus [13].

Given the high degree of interconnectivity between cellular molecular components, it is increasingly evident that a pathology can be characterized by specific modules, or connections of networks between transcriptome, proteome and metabolome. These modules collectively constitute the set of disease-specific clusters of networks, called the pathosome. The connections among different modules are given by the nodes, i.e., proteins interacting with components of different pathways. DDX3X and DDX5 may thus be considered important nodes in the cancer pathosome, whose functions can be influenced by their interacting partners. Based on these considerations, here we will also analyze the protein interaction networks of the two helicases, as reported by the BioGRID and IntAct databases, in different tumors, identifying common and unique interactors with the aim of highlighting novel potential venues of research that might help to explain their apparently conflicting roles as cancer drivers or suppressors [14].

## 2. The DDX3X RNA Helicase

The gene for the DDX3X protein is located on the X-chromosome p11.3–11.23 region that escapes X-inactivation in females and encodes for a 662 amino acid protein. DDX3X is mainly localized in the cytoplasm, but consistent with its numerous associated functions, it has been described to actively shuttle between the nucleus and the cytoplasm [15]. Besides the general roles in the RNA metabolism shared with other DEAD-box proteins, DDX3X has gained particular attention due to its role in viral infections [16] and in the maintenance of genomic stability by playing an active role in the response and repair of DNA damages [10,17]. Moreover, a large body of data published over the years has demonstrated the roles of DDX3X in cancer biology [18]. Alterations in its nucleo-cytoplasmic shuttling and therefore the change of the subcellular localization of DDX3X could have profound effects on its function and might lead to tumorigenesis. However, the genetic background present in particular cell types coupled with a modified expression and localization of DDX3X gives this protein an apparent contradictory dual role in different cancer types [19]. As summarized in Table 1, DDX3X can indeed act as an oncogene and/or a tumor suppressor by altering different molecular pathways that either enhance or block tumor cell proliferation. Remarkably, this dual role of DDX3X has been reported not only in different types of cancer but also in the same cancer (Table 1).

Below, we will briefly summarize the known roles of DDX3X in different cancer types.

### 2.1. Breast Cancer

In breast cancer, DDX3X is overexpressed and acts as an oncogene by promoting proliferation and neoplastic transformation of breast epithelial cells. It inhibits Kruppel-like factor 4 (KLF4) expression by manipulating the KLF4 mRNA alternative splicing. KLF4 is a downregulator of the cell cycle factors CCNA2 and CDK2 and a negative regulator of the G1/S phase transition. DDX3X directly interacts with KLF4 mRNA and regulates its splicing, thus enhancing cell cycle progression. Accordingly, in MCF7 breast cancer cells, the knockdown of DDX3X induces the expression of KLF4, leading to reduced cell growth [20]. Moreover, DDX3X expression in breast cancer was found to be increased by hypoxia-inducible factor 1-alpha (HIF-1α), a transcription factor that induces the transcriptional activation of DDX3X via binding to the HIF-responsive element located in the DDX3X promoter. Hypoxia is a major characteristic of solid tumors. In response to hypoxia, DDX3X is overexpressed in the immortalized breast epithelial cell line MCF10A, and as a consequence, it reduces the expression of E-cadherin via the Rac1-mediated signaling pathway and promotes tumor metastasis [21]. A recent study demonstrated also that DDX3X physically interacts with estrogen receptor-alpha (ERα) and positively regulates its activity as a promoter of cancer progression [22].

### 2.2. Colorectal Cancer

The role of DDX3X in colorectal cancer is controversial. In this carcinoma, DDX3X facilitates tumor metastasis enhancing the expression of the oncogene KRAS by promoting SP1 transcription factor binding to the KRAS promoter. In addition, DDX3X promotes KRAS-dependent tumor invasion by activating the β-catenin/ZEB1 pathway that requires β-catenin/TCF4 activation by DDX3X through the Wnt/CK1ε/Dvl2 axis [24]. In cancer cells harboring wild-type KRAS, increased KRAS induces ROS production, which is followed by increased expression of HIF-1α and YAP1. HIF-1α, in turn, increases DDX3X expression, thus further promoting tumor progression. In addition to its oncogenic role, YAP1 participates in the YAP1/SIX2 pathway involved in the resistance toward anti-EGFR antibody cetuximab treatment in colorectal cancer [23]. However, a different study reported that in colorectal cancer DDX3X is also a tumor suppressor factor. In fact, DDX3X knockdown leads to upregulation of Snail and to a decrease in E-cadherin expression with a consequent reduction in cell aggregation. This suggested that modulation of the Snail/E-cadherin pathway by DDX3X suppresses colorectal cancer cell metastasis [25]. However, it cannot be excluded that the dual role of DDX3X observed in colorectal cancer could possibly be due to the use of different cell lines. Considering the multiple functions of DDX3X in this type of cancer, it is also impossible to predict survival outcomes of patients only using the DDX3X protein as a biomarker, since it seems to associate with different molecules that alter the tumor development.

### 2.3. Lung Cancer

The role of DDX3X in lung cancer is still contradictory; in fact, in lung cancer cells, DDX3X was found overexpressed and was associated with short patient survival. Its oncogenic action is due to the activation of Wnt signaling that promotes tumor progression. In fact, inhibition of DDX3X resulted in growth arrest due to impairment of the Wnt/β-catenin pathway [26]. Moreover, in lung cancer, the long non-coding RNA (lncRNA) LINC00673-v4 is upregulated and associated with disease progression. At the molecular level, this lncRNA promotes the interaction of DDX3X and CK1ε that sequentially stimulates the Wnt/β-catenin signaling [27]. On the other hand, some studies reported an oncosuppressive role of DDX3X. In lung cancer, p53 directly regulates the transcription of DDX3X that in turn enhances the transcription of p21, a well-known cyclin-dependent kinase inhibitor important for cell growth arrest. Indeed, in lung cancer cells, the decreased level of DDX3X due to the loss of p53 transactivating activity is linked with cancer progression [28]. Moreover, in human papillomavirus-positive lung cancer cells, E6 protein suppressed DDX3X expression in a p53-dependent manner (by p53 inactivation). The downregulation of DDX3X suppresses E-cadherin expression by MDM2/Slug pathway expression and promotes tumor metastasis [29].

### 2.4. Hepatocellular Carcinoma

DDX3X overexpression was found in hepatocellular carcinoma (HCC) cells [30]; however, while an oncogenic role of DDX3X in HCC has never been clearly demonstrated, several studies proposed a molecular mechanism that supports an oncosuppressive role of DDX3X. In fact, knockdown of DDX3X upregulates cyclin D1 and downregulates p21, thus promoting cell cycle progression to the S phase and facilitating tumor cell growth [31]. Furthermore, in a p53-independent manner, DDX3X interacts and cooperates with the transcription factor Sp1 to upregulate the promoter activity of p21 to exert its tumor suppressor function [32], and in HepG2 cells, DDX3X promotes the biogenesis of a subset of oncosuppressive miRNAs [33]. Additionally, Rotterlin, a protein kinase inhibitor that exerts its anti-tumor activity in various types of human cancers, was found to upregulate DDX3X expression in HCC that subsequently downregulates cyclin D1 expression and increases p21 level, leading to cell cycle arrest [34].

### 2.5. Prostate Cancer and Ewing Sarcoma

Both prostate cancer and Ewing sarcoma are characterized by an elevated expression of DDX3X. RK-33 is a synthetic DDX3X inhibitor that specifically binds to the ATP-binding cleft of DDX3X, decreasing its unwinding activity. An in vivo combination treatment of RK-33 and radiation obtained a synergistic inhibitory effect on prostate tumor proliferation [35]. Moreover, the inhibition of DDX3X activity by RK-33 altered the Ewing sarcoma cellular proteome, reducing malignant cell growth [36]. However, the molecular mechanism of the oncogenic role of DDX3X in both types of cancers is not completely understood [37].

### 2.6. Oral Squamous Cell Carcinoma and Head and Neck Squamous Cell Carcinoma

A clear mechanism that explains the roles of DDX3X in oral squamous cell carcinoma (OSCC) has not been elucidated yet. In this type of cancer, the levels of DDX3X protein expression are differentially linked to survival or cancer progression, depending on the tumor type. High DDX3X expression is related to poor survival in smoker patients [38]. On the contrary, a low expression of DDX3X is associated with a poor prognosis in non-smoker patients with OSCC [39]. Thus, DDX3X seems to assume a dual role, acting either as a promoter of cancer progression or as a tumor suppressor. A recent study described DDX3X as a regulator of secretor signaling factors in OSCC. In particular, DDX3X was found to mediate the translational control of AREG that in OSCC is an important promoter of cell growth and cell migration through EGFR activation. Six identified missense mutations of DDX3X increased the expression of AREG, suggesting a predominantly oncogenic role in this type of cancer [40]. Instead, in head and neck squamous cell carcinoma (HNSCC), the oncogenic action of DDX3X was recently described. DDX3X was found to form a complex with the cap-binding complex CBC and eIF3 to promote the transcription of ATF4, a potent activator of the epithelial–mesenchymal transformation that facilitates tumor metastasis [41].

### 2.7. Oncogenic Role of DDX3X in Other Cancers

The involvement of DDX3X was also described for other types of cancers where it was mainly investigated as a biomarker for the prediction of survival. Evidence showed that DDX3X is overexpressed in glioblastoma, medulloblastoma, gallbladder carcinoma, pancreatic ductal adenocarcinoma and chronic myeloid leukemia [42,43]. In these cancer types, a high DDX3X expression level is always linked to a poor clinical outcome, confirming the potential oncogenic role in controlling tumor proliferation and progression. In medulloblastoma cell lines, as for prostate cancer and Ewing sarcoma, the treatment with the DDX3X inhibitor RK-33 reduces cell growth by reducing transcript levels of Wnt-regulated genes [44]. Thus, it has been proposed that DDX3X exerts its oncogenic role through the stimulation of the Wnt/β-catenin axis. Consistently, DDX3X mutations that enhance the transactivation capacity of a mutant β-catenin were found in medulloblastoma. The synergic mutation of DDX3X and β-catenin was correlated with an alteration of Wnt signaling which increased cellular proliferation [45]. Furthermore, in medulloblastoma, DDX3X mutants led to impairment in mRNA translation, causing the hyper-assembly of stress granules [46]. Overexpression of DDX3X is a poor prognosis marker in gallbaldder carcinoma and pancreatic ductal ccarcinoma [47,48] and in pancreatic ductal carcinoma, DDX3X promotes p62 accumulation that in turn facilitates epithelial–mesenchymal transition and metastasis [49]. Interestingly, DDX3X mutations that cause alteration of protein function rather than loss of function have been described also in chronic myeloid leukemia [49].

### 2.8. Oncosuppressive Role of DDX3X in Other Cancers

DDX3X is described as a repressor of tumorigenesis in melanoma and NK/T-cell lymphoma. In melanoma, DDX3X promotes the mRNA translation of MIFT, a regulator of development in various cell types that is correlated with less invasive tumors when highly expressed. However, DDX3X activity is frequently lost in human metastatic melanomas. As a consequence, the loss of DDX3X altered MITF translational regulation, triggering a proliferative-to-metastatic phenotypic switch in melanoma cells [50]. It was also reported that DDX3X escapes X chromosome inactivation and is preferentially mutated in male melanoma patients, supporting the sex differences observed in this tumor [51]. DDX3X mutations are normally present in NK/T-cell lymphoma and are correlated with a poor clinical outcome. Most of them affect the two highly conserved RecA-like domains, producing truncated DDX3X variants with a decreased helicase unwinding activity. This loss of function of DDX3X abnormally activates NF-κB and MAPK pathways at the transcriptional level and is associated with an impaired ability to suppress cell cycle progression [52]. DDX3X mutated proteins are also connected to an increase in STAT3/p42/p44 phosphorylation and to the development of chemoresistance [53]. Finally, high levels of DDX3X expression in gastric cancer are correlated with a better prognosis and patient survival [42].

## 3. The DDX5 RNA Helicase

DEAD-box polypeptide 5 (DDX5) was first identified in the 1980s. It is also named p68 after its molecular weight (MW) of about 68 kDa. Its gene encodes a 614 aa protein and it is located on chromosome 17q23 [54]. DDX5 has orthologs in various organisms, from *Escherichia coli* to humans [55].

The DDX5 helicase is a multifunctional protein involved in many processes. Similarly to DDX3X, it participates in several cellular RNA transactions, from transcription, mRNA splicing, export and translation to ribosome biogenesis and miRNA processing. Through these multiple activities, DDX5 is implicated in cell cycle regulation and apoptosis [56]. Not unexpectedly, alterations in DDX5 metabolism are linked to tumorigenesis, even though the mechanisms underlying the DDX5 oncogenic role may differ from one type of tumor to another and, in some tumors, are not fully characterized. An important role seems to be played by post-translational modifications. For example, the phosphorylation of DDX5 residues Y593 and Y595 (tyrosine) has been shown to increase cell proliferation, EMT and metastases, as well as suppressing TRAIL-induced apoptosis [57,58,59]. Conversely, phosphorylation of T564 and T446 residues promotes apoptosis of cancer cells and potentiates the effects of anticancer drugs such as oxaliplatin [60]. DDX5, similarly to DDX3X, has been also implicated in the maintenance of genome stability, participating in DNA damage response [10]. Overall, the abnormal expression of DDX5 in many types of cancer and its regulation support an important role of this protein in promoting cancer cell proliferation [61].

### 3.1. DDX5: Oncogene or Suppressor?

The available data on DDX5 indicated that it acts much more often as an oncogene, but it can also have tumor suppressor functions in specific cancer types (Table 2). The roles of DDX5 in different cancer types are detailed in the next sections.

### 3.2. Colon and Colorectal Cancer

In colon cancer, it was reported that overexpression of DDX5 upregulates the levels of the AKT protein and mRNA, increasing the transcription of the AKT gene promoter by DDX5 itself, β-catenin and NF-κB [62,63]. The increase in AKT transcription coupled with the depletion of the tumor suppressor FOXO3a (a downstream gene) accounts for the oncogenic role of DDX5 in colon cancer. In addition, DDX5 also regulates the expression of RelA, a component of the NF-κB signaling pathway, which is involved in colon tumorigenesis [64]. In colorectal cancer (CRC), the oncogenic function of DDX5 has been observed in various studies. DDX5 directly interacts with O-linked N-acetylglucosamine transferase (OGT) which stabilizes the DDX5 protein through O-GlcNAcylation, resulting in high expression levels. DDX5, in turn, promotes the phosphorylation of mTOR by activating the AKT pathway, so DDX5 exerts its oncogenic function in CRC by mediating the cross-talk between O-GlcNAcylation and AKT/mTOR signaling [65]. In addition, DDX5 has been found to associate with fructose biphosphate aldolase A (AldoA), an aldolase isoenzyme involved in glucose metabolism and highly expressed in different types of tumors. AldoA and DDX5 were overexpressed in both primary and metastatic CRC of the liver, with respect to normal tissues of the glandular epithelium. While the functional significance of the interaction between AldoA and DDX5 is not well characterized, their concomitant overexpression correlates with poor prognosis in CRC patients [66].

### 3.3. Breast Cancer

DDX5 expression has been found to increase progressively from luminal to basal cells in breast cancer, with a concomitant expression of CD44 [67,68]. DDX5 has been found to regulate the biogenesis of a subset of miRNAs, targeting different downstream mRNAs (e.g., PDCD4 via miR-21, cofilin and profilin via miR-182) and thus promoting the progression of breast cancer. In early-stage breast cancer, DDX5 expression along with overexpression of EGFR and Ki67, a nuclear marker for cell proliferation, positively correlates with a poor prognosis and invasiveness of tumors, a high risk of recurrence and worse survival in patients [68]. DDX5 is also a co-activator of β-catenin, and it has been shown to be involved in Wnt pathway-mediated expression of TCF4 in breast cancer cells. Thus, DDX5 seems to govern the assembly of a transcription activation complex, which regulates the expression of TCF4. Therefore, the regulation of the DDX5 gene represents an important mechanism for the control of proliferation mediated by Wnt signaling in tumorigenesis [69]. These data indicate that DDX5 acts as an oncogene in breast cancer.

### 3.4. Leukemia

Inhibition of DDX5 expression in acute myeloid leukemia (AML) significantly reduces cancer cell proliferation in vitro and progression in vivo [70]. Consistent with these results, depletion induces apoptosis of leukemic cells through the stimulation of genes regulated by the NOTCH1 pathway [71]. One study analyzed the effects of the anti-DDX5 monoclonal antibody 2F5 on acute promyelocytic leukemia (APL). Treatment with DDX5-targeting 2F5 or anti-DDX5 siRNA could inhibit APL cell proliferation by downregulation of DDX5, inducing G0/G1 phase arrest. Interestingly, unlike previous data related to DDX5, it was found that 2F5 targeted at DDX5 has no effect on the proliferation of acute T-lymphocytic leukemia (T-ALL) cell lines (Jurkat and CEM-C7). The results obtained showed that the levels of basal expression of DDX5 in different leukemia cell lines were also different; the baseline level of DDX5 in APL cell lines was much higher than that in the T-ALL cell line. These results indicated that the different baseline levels of DDX5 could determine the susceptibility of different leukemia subtypes to 2F5 [72].

### 3.5. Lung Cancer

DDX5 was found overexpressed in non-small-cell lung cancer (NSCLC) tissues compared to adjacent normal matched tissues. DDX5 overexpression correlated with advanced clinical stage, a high proliferation index, and worse overall survival in NSCLC patients. Consistently, overexpression of DDX5 promoted NSCLC cell proliferation in vitro and in NSCLC xenotransplants in vivo, while downregulation of DDX5 showed an opposite effect. DDX5 activated the expression of cyclin D1 and c-Myc through its interaction with β-catenin which promoted its nuclear translocation. As expected, β-catenin silencing abrogated the DDX5-dependent expression of cyclin D1 and c-Myc and proliferation in NSCLC cells [73].

Interestingly, DDX5 depletion in small cell lung cancer (SCLC) inhibited the TCA cycle by reducing intracellular succinate (a direct electron donor to the mitochondrial complex II). Thus, DDX5 downregulation resulted in reduced mitochondrial growth, leading to various metabolic dysfunctions in SCLC cells [74].

### 3.6. Osteosarcoma

Osteosarcoma (OS) is a bone malignancy with no currently available effective treatments. Since lncRNAs have been found to participate in OS progression, the involvement of lncRNA DLEU1, which acts as an oncogene in different cancers, was recently explored. In OS cells, downregulation of DLEU1 inhibits cell proliferation, migration and invasion, confirming its oncogenic role. DLEU1 exerts its function by regulating the expression of DDX5 through the miRNA miR-671-5p. Normal levels of miR-671-5p downregulate DDX5 and prevent tumor progression. Instead, by directly targeting miR-671-5p, DLEU1 increases the expression of DDX5, facilitating the proliferation of OS cells. This interaction indicates that high levels of DDX5 are associated with tumor development and suggests the oncogenic role of DDX5 in OS [75].

### 3.7. Prostate Cancer

It has been shown that in prostate cancer (PC) DDX5 is highly expressed and promotes the proliferation of cancer cells [56]. Some studies identified DDX5 as a novel activator of androgen receptor (AR) and β-catenin transcription in PC, and therefore DDX5 could play a role in the progression of PC to hormone-refractory disease. It was also found that the oncogenic lncRNA CCAT1 promotes PC cell proliferation and tumor growth of PC xenotransplantation, acting as a scaffold for the DDX5 and AR transcriptional complex to facilitate the expression of AR-regulated genes in nuclei, thus stimulating PC progression [76]. In addition, another study on PC patients found DDX5 fused in frame with the ETV4 transcription factor, leading to the expression of a DDX5-ETV4 fusion protein. While the functional significance of this fusion is not known, ETV4 belongs to the family of E26 transformation-specific transcription factors, which are known to often undergo fusions in cancer cells and promote cellular proliferation, carcinogenesis and metastasis [77].

### 3.8. Gastric Cancer

In gastric cancer, DDX5 induces tumor proliferation by activating the mTOR/S6K1 pathway. Consistently, the mTOR inhibitor everolimus significantly reduced DDX5-mediated cell proliferation [78]. In addition, tissues and cell lines of gastric cancer overexpress the lncRNA MIAT which sequesters the miRNA miR-141, a negative regulator of DDX5 expression. Thus, high levels of MIAT result in inhibition of miR-141 synthesis and increased levels of DDX5, promoting cell proliferation and metastasis in gastric cancer [79].

### 3.9. Glioblastoma

In glioblastoma, DDX5 has been observed to bind the N-terminus of the NF-κB p50 subunit, increasing the transcriptional activity of the p50 target genes and stimulating the growth of glioma cells [80]. DDX5 has been identified as a protein that interacts with the lncRNA LINC01116, both by Western blot and RIP assays. LINC01116 is overexpressed in gliomas and could recruit DDX5 to the IL-1β promoter region to activate its transcription. This, in turn, facilitates infiltration by tumor-associated neutrophils, promoting an inflammatory microenvironment that stimulates tumor growth [81]. It was also shown that DDX5 is a negative regulator of the dual specificity phosphatase DUSP5. DUSP5 negatively regulates the ERK proliferation pathway; thus, overexpression of DDX5 observed in gliomas causes hyperactivation of ERK through downregulation of DUSP5, promoting cancer proliferation [82].

### 3.10. Cervical Cancer

The levels of DDX5 mRNA and protein were shown to be significantly increased in cervical cancer cell lines (CaSki, HeLa HPV18-positive, SiHa HPV16-positive and C-33A HPV-negative) compared to the HaCaT cell line of human keratinocytes; DDX5 overexpression greatly improved the migration capacity of CaSki cells through the activation of TGF1-β pathway, with an increase in epithelial–mesenchymal transition markers and alterations of cell morphology [83].

### 3.11. Endometrial Cancer

Expression of hepatoma-derived growth factor (HDGF) and DDX5 were positively correlated in endometrial cancer (EC) tissues. HDGF promotes β-catenin expression by functioning as a transcriptional activator of its promoter. DDX5 forms a complex with β-catenin, increasing its transcription activation activity. In turn, β-catenin functions as a transcriptional activator of DDX5 gene expression by binding to its promoter. Thus, HDGF can upregulate DDX5 levels by inducing β-catenin. Indeed, transfection of EC cells silenced for HDGF with the cDNA of β-catenin significantly increased DDX5 expression [84]. These data suggested a cooperative role of HDGF and DDX5 in inducing β-catenin, promoting cell growth and metastases in the EC.

### 3.12. Squamous Cell Carcinoma

Studies relevant to SCC also support the idea that DDX5 is important for tumor proliferation. Differential visualization and Northern blots found that DDX5 was overexpressed in head and neck SCC cells (HNSCCs) but not in normal keratinocytes of the mucous membrane of the upper aerodigestive tract. This led to the hypothesis that DDX5 may promote malignant transformation or progression of HNSCCs [85]. In another study, DDX5 silencing in esophageal SCC cells (ESCCs) suppressed CDK2, cyclin D1 and vimentin expression while upregulating E-cadherin and inhibiting cell proliferation and metastases. Thus, reduced levels of DDX5 in ESSCs seem to be related to lower malignancy [86].

### 3.13. Human Hepatocellular Carcinoma

An oncosuppressive activity of DDX5 has been observed so far only in hepatitis B virus (HBV)-associated human hepatocellular carcinoma (HCC). DDX5 interacts with the autophagic receptor p62, promoting autophagy and suppressing tumorigenesis. Indeed, in HCC, whether HBV-associated or not, DDX5 is inversely correlated with p62/sequestrosome 1 (SQSTM1) expression [87]. Another significant contribution to the development and progression of cancer is the aberrant expression of miRNAs. DDX5 interacts with the Drosha microprocessor complex, regulating the biogenesis of miRNAs [88,89]. In HCC, repression of DDX5 has been proposed to cause a decrease in miRNA biogenesis, which is linked to hepatocarcinogenesis. Indeed, low DDX5 expression correlated with a poor prognosis in patients after tumor resection [90]. In another study, DDX5 was found to interact with SUZ12, the core subunit of the chromatin-modifying PRC2 complex, and, in association with the lncRNA HOTAIR, to repress transcription of the cancer stem cell marker EpCAM, as well as that of stemness-related genes Nanog, Oct4 and Sox2. These results showed, for the first time, a role of DDX5 in transcriptional repression of specific cellular genes, together with HOTAIR and the PCR2 complex [91].

## 4. The DDX3X and DDX5 Interactome

The specific roles of DDX3X and DDX5 in regulating cell proliferation largely depend on the network of interactions with different factors in different metabolic pathways. Both DDX3X and DDX5 are characterized by a large interactome, with hundreds of identified interacting proteins. However, in many cases, the functional significance of such interactions is not known. With the aim of providing a functional overview of the DDX3X and DDX5 interactomes, identifying both common and unique pathways, protein interactors of DDX3X and DDX5 were downloaded from BioGrid https://thebiogrid.org (accessed on 14 March 2022) and IntAct https://www.ebi.ac.uk/intact/home (accessed on 14 March 2022). These databases are public repositories that collect and make available data on protein interactions from model organisms. Both databases provide manually curated datasets of protein interactions derived from high-throughput analyses and from the literature; protein interaction data refer to a wide range of cellular and tissue systems and are not restricted to any pathological condition. Both databases were searched using the name of the two genes, and searches were restricted to human data.

The lists of interactors obtained from the two databases were then merged to obtain non-redundant lists of 551 and 410 interactors for DDX3X and DDX5, respectively (Appendix A Appendix A). The intersection of the two lists highlighted the existence of 193 interactors common to the two proteins and 357 and 217 interactors specific for DDX3X and DDX5, respectively (Figure 2A). In order to characterize the DDX3X and DDX5 interactome, the three lists were functionally annotated with respect to the Gene Ontology (GO) Biological Process categories using the DAVID tool (https://david.ncifcrf.gov/tools.jsp, accessed on 8 April 2022). A total of 1091, 801 and 494 GO terms were found significantly enriched (FDR < 0.05), respectively, in the lists of common, DDX3X-specific and DDX5-specific interactors. Using the Functional Clustering function of DAVID, GO terms were grouped to shed light on the most enriched biological processes of the common and specific pathways. In Figure 2B, the top five GO terms (i.e., the most representative for each enriched cluster) are represented.

To evaluate the putative involvement of DDX3X and DDX5 interactors in specific types of cancer, a functional annotation was performed with respect to the DisGeNET (https://www.disgenet.org/, accessed on 8 April 2022) and the GAD (https://maayanlab.cloud/Harmonizome/dataset/GAD+Gene-Disease+Associations, accessed on 8 April 2022) databases implemented in DAVID. As shown in Figure 2B, different functional classes were found most enriched by GO analysis for the specific vs. common interactors of DDX3X and DDX5, highlighting how important the nature of the interactome is in mediating the different functions of these helicases. DDX3X and DDX5 common interactors, as well as the specific ones, annotated with respect to different types of tumors, were used for further analyses.

### 4.1. DDX3X and DDX5 Share Common Interactors with Known Roles in Tumorigenesis

By analyzing the BioGrid and IntAct databases, we have identified interactors common to DDX3X and DDX5. Below (Table 3) we selected those interactors based on the information obtained through the two databases of interest, for which robust literature data were available to support their roles in cancer progression, according to the type of tumor in which they act.

To the best of our knowledge, no information on the functional significance of these interactions has been reported in the literature. However, by comparing the known roles of the identified interactors with those of DDX3X and DDX5 in the same cancer types, some observations can be made. For example, in breast cancer, both DDX3X and DDX5 act as oncogenes. On the other hand, among the eight common interactors identified in this type of cancer, six were oncogenes and two were oncosuppressors. Thus, it might be hypothesized that DDX3X and DDX5 exert their tumor-promoting functions in breast cancer by enhancing the expression/activity of oncogenes and repressing oncosuppressors. In lung cancer, both DDX3X and DDX5 have been found to physically interact with the oncogene MATR3. However, while DDX5 acts as an oncogene, thus presumably enhancing the tumor transformation function of MATR3, DDX3X can act both as an oncogene and as an oncosuppressor. It would thus be interesting to investigate the functional significance of the MATR3–DDX3X interaction, in order to determine which of those two opposing roles of DDX3X might be influenced by this protein. A special case is represented by the transcription factor YY1, which was found to interact with both DDX3X and DDX5 in cervical cancer, where it can act both as an oncosuppressor and an oncogene, depending on the subset of genes that are activated. DDX5 acts as an oncogene in cervical cancer, which raises the question of whether it exerts its tumor-promoting function by enhancing the oncogenic activity of YY1 or repressing its oncosuppressive role. The role of DDX3X in this kind of cancer has not been assessed, but it has been shown that in gastric, prostate and colon cancer cell lines, the interaction of DDX3X with YY1 is mediated by the circular DNA circ-CTNNB1, promoting tumor proliferation through the deregulation of the β-catenin axis [117]. Thus, it would be worth investigating whether this mechanism plays a role in modulating the functions of DDX3X and YY1 in cervical cancer.

### 4.2. Unique Interactors of DDX3X

Based on literature data, several DDX3X unique interactors were found associated with different cancer types. No functional characterization of these interactions was available, to the best of our knowledge, at the time of writing. Thus, the possible significance of their functional relationships with DDX3X as discussed below can only be in the form of working hypotheses, which we propose as starting points for deeper experimental investigations.

#### 4.2.1. Breast Cancer

Alpha-smooth muscle actin (ACTA2) is involved in tumor progression and metastasis. Dimerization of EGFR and HER2 receptors mediates ACTA2 overexpression through the JAK2/STAT1-dependent signaling pathway to trigger motility of breast cancer cells [118]. A functional relationship between ACTA2 and DDX3X has been described in HNSCC, where DDX3X increased the translation of ATF4 that upregulated different proteins, including ACTA2, which promoted epithelial–mesenchymal transition (EMT) [40]. Thus, the interaction between DDX3X and ACTA2 in breast cancer might be related to the same mechanism and could be further explored.

The G3BP2 (Ras GTPase-activating protein-binding protein 2) proteins play an essential role in the formation of the stress granules, a mechanism that is thought to protect cancer cells from apoptosis or induce resistance to radiation or anticancer drug treatments [119]. In breast cancer, G3BP2 regulates tumor initiation through the stabilization of SART3 (squamous cell carcinoma antigen recognized by T cells 3) mRNA [120]. DDX3X is involved in stress granule formation, and DDX3X mutants were shown to induce the hyper-assembly of stress granules in medulloblastoma [49]. Hence, it is possible that the DDX3X protein could enhance the effect of G3BP2 in breast cancer and play an oncogenic role in tumor initiation.

PABPC1 (poly(A)-binding protein cytoplasmic 1) regulates initiation of translation and mRNA decay by binding to the poly(A) tail of mRNA. In breast cancer, the long non-coding RNA SNHG14 promotes cancer progression, upregulating PABPC1 expression [121]. It was already demonstrated that the DDX3X helicase co-localizes and physically interacts with PABPC1 in fibroblasts, promoting cell migration and spreading [122]. However, in breast cancer, the role of their interaction has not been characterized yet. PABPC1 is highly expressed in hepatocellular carcinoma and ovarian cancer [123,124]. While the role of DDX3X in hepatocellular carcinoma is mainly as a tumor suppressor, its involvement in ovarian cancer is still unknown. Investigating the interplay of DDX3X with PABPC1 could better explain the role of DDX3X in liver cancer as well as define its implication in ovarian cancer.

FBXW7 (F-box and WD repeat domain containing 7), a member of the F-box protein family, is a regulator of proteasome-mediated degradation of several proteins, including oncoproteins such as cyclin E, c-Myc, Mcl-1, mTOR, Jun, NOTCH and AURKA. Accordingly, FBXW7 is a critical tumor suppressor and one of the most frequently mutated proteins in human cancers. Eighty percent of breast cancer cell lines contain alternatively spliced isoforms of FBXW7 and therefore a lower expression of the wild-type protein associated with a higher expression of cyclin E1 protein, a marker of proliferation [125]. DDX3X is a regulator of cell cycle progression through translational control of cyclin E1. Knockdown of DDX3X results in a reduction in cyclin E1 causing cell cycle arrest in the G1 phase in breast, lung, colorectal and prostate cancers and in medulloblastoma [126]. As a known regulator of mRNA splicing, export and translation, DDX3X could exert its oncogenic role by downregulating FBXW7 mRNA translation by modifying its alternative splicing.

Other DDX3X unique interactors were BTK2 (Bruton tyrosine kinase), FXR1 (fragile X-related protein-1), NEDD4 (E3 ubiquitin-protein ligase) and NOS2 (nitric oxide synthase 2) genes. These proteins, when overexpressed, promote breast tumor metastasis [127,128,129,130]. However, a common pathway that includes DDX3X is still undefined.

#### 4.2.2. Colorectal Cancer

Some unique interactors of DDX3X are associated with colorectal cancer. Since in this type of tumor the positive or negative role of DDX3X is not completely clear, the possible connection with these proteins remains elusive.

ICAM-1 (intracellular adhesion molecule-1), when phosphorylated by the protein kinase c-MET, can interact with the Src kinase to increase its activity, promoting colorectal cancer progression [131,132]. It could be interesting to understand whether DDX3X affects ICAM-1 phosphorylation.

S100A9 (S100 calcium-binding protein A9) is a calcium-binding protein involved in the regulation of inflammation and immune response. Its overexpression in colorectal cancer has been reported to upregulate the activity of the Wnt/β-catenin pathway, promoting cancer cell migration [133]. DDX3X is a central modulator of the Wnt/β-catenin pathway. Thus, both proteins act as oncogenes by sharing the same pathway, suggesting a possible connection that must be elucidated.

#### 4.2.3. Oral Squamous Cell Carcinoma

Our analysis identified the specific interaction of DDX3X with different members of the KRT family involved in oral squamous cell carcinoma (OSCC) proliferation, such as KRT5, KRT14 and KRT17 [134,135,136]. KRT17 stimulates the ATK/mTOR pathway and upregulates the expression of SLC2A1 (solute carrier family 2 member 1) for glucose uptake to induce cell proliferation and migration [135]. DDX3X has already been described as an activator of AKT in colorectal cancer, promoting metastasis via the β-catenin/ZEB1 axis [24]. Thus, in OSCC, the interaction of DDX3X with KRT17 could have an enhancer effect in the development of tumorigenesis.

Instead, KRT5 and KRT14 are positive mediators of cell differentiation and neoplastic transformation whose expression level is regulated by TAP63 and NOTCH1 [136]. To date, there are no clear reports of DDX3X interaction with TAP63 or NOTCH1. A possible connection network that also includes DDX3X in the regulation of KRT5 and KRT14 could be studied.

Our analysis also identified KRT1, KRT6b and KRT10 as interactors of DDX3X. These proteins are overexpressed in squamous cells of carcinoma of the oral cavity, and they have a possible association with the p21 signaling pathway [137]. In lung cancer, DDX3X acts as an oncosuppressor by increasing the production of p21, a tumor suppressor protein, through the promotion of p53/SP1 interaction [28]. Considering that a high expression level of these keratins is associated with low survival in patients with melanoma [138], another tumor where DDX3X acts as a negative regulator of proliferation [50], it seems that KRT1, KRT6b and KRT10 and DDX3X could have a sort of antagonist interaction in the regulation of the expression of p21. This possibility may also explain the dual role of DDX3X described in the literature for OSCC.

#### 4.2.4. Other Cancers

The interactome analysis of DDX3X highlighted its interaction with several miRNAs that could have an important role in the development of various types of cancer. As reported in the literature, DDX3X can interact with the Drosha/DGCR8 complex and stimulate its pri-miRNAs processing activity, thus enhancing mature miRNA expression levels [139]. The involvement of DDX3X in miRNA biogenesis has been suggested for 14 types of cancer, including breast, colorectal, liver, squamous cell carcinoma, bladder, ovarian and acute myeloid leukemia, which were also found in our analysis. The miRNA regulation by DDX3X can target the expression of multiple downstream genes and impact its oncogenic or oncosuppressive role [33].

Besides miRNA biogenesis regulation, the role of DDX3X ovarian cancer has not been defined yet. Among DDX3X unique interactors, we identified ISG15 (interferon-stimulated gene 15), which in ovarian cancer is involved in tumor suppression by decreasing the expression levels of phospho-ERK1 and ERK downstream genes that are associated with cancer progression [140]. It was reported that DDX3X during viral infection can regulate the production of interferon type-1 and of interferon-stimulated genes, including ISG15 [141]. Moreover, ISG15 is a negative regulator of the NF-κB pathway [142,143]. In ovarian cancer, tumor progression is determined by the activation of non-canonical NF-κB signaling that has NFκB2 as a major component [144,145]. NFκB2-transcription-mediated activity can be suppressed by the interaction of DDX3X with NF-κB [146]. Therefore, is possible that DDX3X exerts an oncosuppressive role in ovarian cancer by acting on the negative regulation of NFκB2 at different levels by direct interaction with NF-κB or by enhancing the expression of ISG15 that in turn suppresses NFκB2 activity. This function could be similar to that already reported in NK/T-cell lymphoma where a loss of DDX3X was associated with a higher expression of NF-κB [52].

### 4.3. Unique Interactors of DDX5

As for DDX3X, several DDX5 unique interactors were associated with different tumor types, but again there was a lack of information about the functional significance of these interactions. However, based on the available data, it is possible to propose educated hypotheses to guide future studies.

#### 4.3.1. Breast Cancer

AKAP8 (A-kinase anchor protein 8), also known as AKAP95, is an RNA-binding protein and a splicing regulatory factor that acts as a tumor suppressor protein by interacting with hnRNPM, antagonizing its splicing activity on CD44 exon skipping [147]. In mammalian cells, AKAP8 is mainly present in the nuclear matrix where it co-localizes and physically interacts with DDX5 [148]. Considering the oncogenic role of DDX5 in breast cancer, the overexpression of DDX5 could be crucial for limiting the oncosuppressive action of AKAP8 through its nuclear sequestration. Indeed, as demonstrated, cells with loss of AKAP8 exhibit accelerated EMT and elevated breast cancer metastatic potential [147].

Our analysis found that GOLIM4 (Golgi integral membrane protein 4) is a specific interactor of DDX5. GOLIM4 acts as a tumor suppressor, but its expression is downregulated by the oncogenic miRNA miR-105-3p which is overexpressed in cancer cells [149]. As already mentioned, DDX5 controls, either directly or indirectly, miRNA maturation. Thus, a high level of DDX5 could either directly inhibit GOLIM4 via direct interaction or enhance the concentration of miR-105-3p to block the suppressor activity of GOLIM4.

KIF23 (kinesin family member 23) exerts a pro-tumor function in breast cancer by stimulating the Wnt/β-catenin pathway [150]. In colorectal cancer, this signaling is activated by the interaction of NEAT1 (nuclear enriched abundant transcript 1) with DDX5 to promote cancer and metastasis [151]. The identification of KIF23 as a direct interactor of DDX5 might suggest that a similar mechanism could be present in breast cancer cells.

MCM5 (minichromosome maintenance complex component 5) has been recently found overexpressed in breast cancer patients [152]. A previous study reported the observation that MCM2, MCM5 and CDC45 proteins are all downregulated in DDX5-depleted breast cancer cells [153]; however, the role of the direct interaction between DDX5 and MCM5 in the regulation of MCM5 levels is presently unclear.

MYOD1 (myogenic differentiation 1) is a transcription factor that inhibits breast cancer differentiation, acting as a tumor suppressor [154]. Since DDX5 acts as an oncogene in breast cancer, the significance of the interaction with DDX5 is unclear, since it was reported that the RNA helicases DDX5 and DDX17 are positive regulators of the transcription activity of MYOD1 [155]. However, the mechanism underlying the oncosuppressor role of MYOD1 in breast cancer has not been discovered yet, so investigating its interaction with DDX5 might provide some clues about the molecular pathways involved.

Another unique interactor of DDX5 is the splicing factor SRSF1 (serine and arginine-rich splicing factor 1). It is upregulated in breast tumors, and its overexpression promotes transformation by the P-AKT/C-MYC axis [156]. Nevertheless, it was demonstrated that the loss of SRSF1 (also known as ASF/SF2) induces R-loop accumulation and genomic instability [157]. SRSF1 interacts with Thrap3 (thyroid hormone receptor associated protein 3) and DDX5, which promote R-loop resolution in breast cancer cells [158]. Thus, the oncogenic role of SRSF1 and DDX5 might also be to increase the DNA damage tolerance of cancer cells by reducing R-loop accumulation and preventing cell death.

#### 4.3.2. Prostate Cancer

MBNL1 (muscleblind-like splicing regulator 1) is a tumor suppressor factor that was found downregulated in different cancers, including prostate cancer [159,160]. Its interaction with DDX5 has been already shown in myotonic dystrophy type I, where DDX5 acts as a modifier of MBNL1 splicing activity [161]. Thus, DDX5 overexpression in prostate cancer might lead to the inhibition of the oncosuppressor activity of MBNL1.

RBFOX2 (RNA binding fox-1 homolog 2) is an important regulator of mesenchymal tissue-specific splicing, implicated in cell differentiation and development. Its deregulation is associated with cancer progression and metastasis [162]. It was found that nuclear RBFOX proteins are part of large multiprotein splicing complexes also containing DDX5 [163]. As for MBNL1, the overexpression of DDX5 in prostate cancer could play a crucial role in the deregulation of the activity of RBFOX2 leading to cancer progression.

#### 4.3.3. Leukemia

Another unique DDX5 interactor is ZFP36L2 (ZFP36 ring finger protein like 2). In leukemia cells, overexpression of ZFP36L2 could promote cell cycle arrest in the G0/G1 phase and induce apoptosis, inhibiting cell proliferation [164]. Moreover, in colorectal cancer, the expression ZFP36L2 inhibits the production of cyclin D, suppressing cell proliferation [165]. DDX5 promotes cyclin D expression in non-small-cell lung cancer through the β-catenin axis [73]. Thus, its direct interaction with ZFPL36L2 might be part of a mechanism whereby DDX5 inhibits the oncosuppressive role of ZFP36L2, further promoting cyclin D expression. A possible additional element of this pathway could be miR-375. DDX5 is essential for the maturation of miR-375 [166] which in turn is a negative regulator of ZFP36L2 expression [167].

SBDS (SBDS ribosome maturation factor) is the protein responsible for Shwachman–Bodian–Diamond syndrome. In acute myeloid leukemia, SBDS is overexpressed and is a direct inhibitor of PP2A (protein phosphatase 2). PP2A inhibition is required for the activation of oncogenic signaling pathways leading to anti-apoptotic processes and cell transformation [168]. Interestingly, DDX5 can directly interact with the catalytic subunit beta of PP2A to inhibit type-I interferon production [169]. Thus, it is possible that DDX5 might mediate the association of SBDS with PP2A, leading to its inhibition and promoting tumor progression in myeloid leukemia cells.

#### 4.3.4. Hepatocellular Carcinoma HBV-Related

In the context of HBV-induced hepatocellular carcinoma, we found ANLN (anillin actin-binding protein) as a DDX5-specific interactor. During HBV infection, the production of miRNAs miR-15a and miR-16-1 is inhibited; both are ANLN repressors. In normal conditions, ANLN inhibition results in activation of apoptosis signaling and DNA damage checkpoints, while high ANLN expression promotes tumor growth [170]. It was reported that p53 can interact with DDX5 and DDX17 (associated with the Drosha complex) to upregulate the expression of a subset of miRNAs, including miR-15a and miR-16-1, to decrease the rate of cell proliferation [171]. Thus, the interaction of ANLN with DDX5 might be an additional component of this pathway and could explain the oncosuppressive role of DDX5 in this type of cancer.

#### 4.3.5. Glioblastoma

HEY1 (Hes-related family BHLH transcription factor with YRPW motif 1) is a transcription factor found upregulated in glioblastoma. Its overexpression is triggered by NOTCH1 and E2F pathways [172]. Since it was already shown that the oncogenic effect of high DDX5 levels also involves the deregulation of NOTCH1 [71] and E2F [153] pathways, it should be interesting to study the functional significance of the interaction of DDX5 with HEY1 in this type of cancer.

## 5. Conclusions

The DEAD-box RNA helicases DDX3X and DDX5 are at the center of many different signal transduction and metabolic pathways, thus qualifying as important nodes of the cellular interactome. As outlined above, many of these pathways are involved in different types of cancer, and DDX3X and DDX5 play a major role in their regulation. The complexity of these interactions is also reflected by the dual role played by these proteins, which, depending on the cell-specific interactome, can act either as oncogenes or oncosuppressors.

Through data mining and bioinformatic analysis, we have provided here an atlas of non-redundant protein–protein interactions centered on DDX3X and DDX5, identifying both common and unique interactors. Based on GO classification we have grouped these interactors according to their association with different cancers. To the best of our knowledge, the functional significance of the vast majority of these interactions has not been investigated yet. By comparing the known cancer-specific roles of the different interactors with the available data for the oncogenic vs. tumor suppressor roles of DDX3X and DDX5, we were able to put forward several working hypotheses which might contribute to explaining the dual role played by these RNA helicases in the different tumors.

For example, as discussed above, several of the unique interactors of DDX3X and DDX5 operate in the cross-talk between the two important signal transduction pathways Wnt/β-catenin and AKT/mTOR. The FBXW7 interactor of DDX3X is a common negative regulator of both mTOR and β-catenin [125,173], while PP2A which interacts with DDX5 is a shared component of both pathways [174]. Similarly, the DDX3X interactor KRT17 upregulates both AKT/mTOR and Wnt/β-catenin signaling [134,135,175]. KIF23 interacts with DDX5 and is a common regulator of both AKT and Wnt/β-catenin pathways [150,176]. DDX3X and DDX5 have been shown to operate in these pathways as well, with either positive or negative regulatory roles depending on the cancer type. Thus, their functional interactions with shared components of these two pathways might differentially affect the proliferative signals, depending on the cancer-specific genetic background.

Another relevant pathway is miRNA biogenesis. As outlined above, both DDX3X and DDX5 are involved in the regulation of the maturation of a subset of miRNAs that play different roles in carcinogenesis [68,139]. Thus, the differential expression or regulation of these two RNA helicases in different tumors might alter the balance of specific miRNAs, influencing tumor proliferation in either a positive or negative way.

In conclusion, we believe that the analysis presented here supports the roles of DDX3X and DDX5 as important nodes in the cancer pathosome and might suggest new venues of research for a better understanding of their roles in tumor transformation and cancer progression.

## Figures and Tables

**Figure 1 cancers-14-03820-f001:**
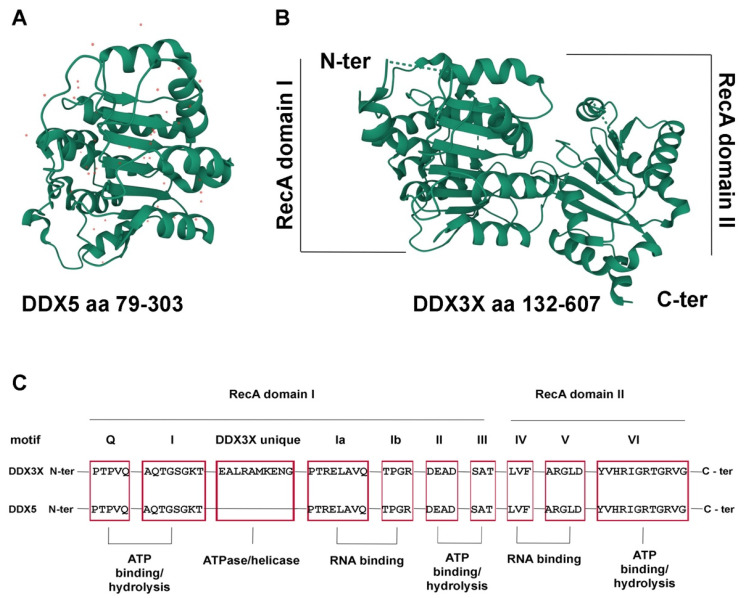
The DDX3X and DDX5 structure. (**A**) Structure of the DDX5 N-terminal core domain (aa 79-303), PDB 4a4d [3]. Red dots represent water molecules. (**B**) Structure of the DDX3X helicase core (aa 132-607), PDB 5E7I [4]. The N-terminal domains of DDX5 and DDX3X are shown in the same orientation to highlight the conserved RecA fold of both proteins. Images were generated by the authors with the online tool Mol*3D viewer (https://www.rcsb.org/3d-view, accessed on 1 August 2022) [5]. (**C**) Schematic representation of the conserved helicase motifs of DDX3X and DDX5 (red boxes) with the respective functional roles.

**Figure 2 cancers-14-03820-f002:**
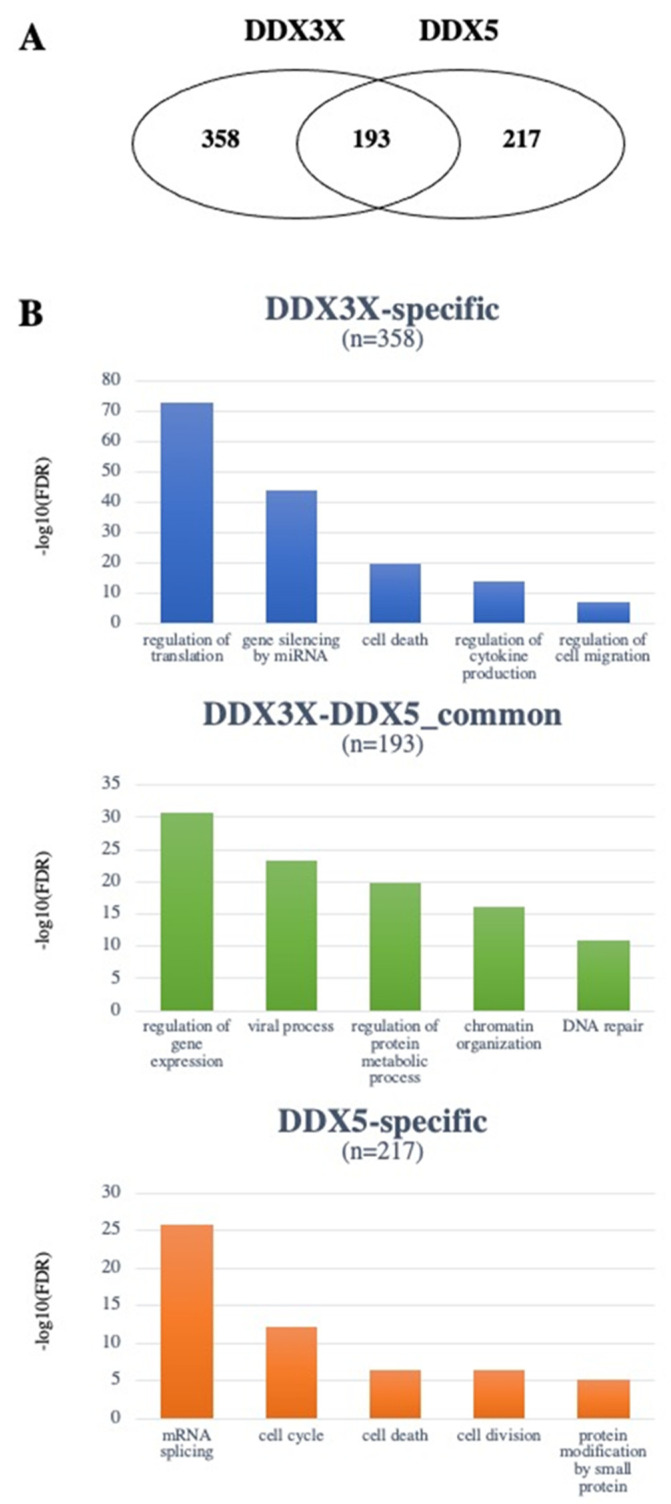
The DDX3X and DDX5 interactomes. (**A**) Venn diagram showing the number of non-redundant common or unique interactors of DDX3X and DDX5. (**B**) GO classification of the most represented functional classes among either common or unique interactors of DDX3X and DDX5. Different *y*-axis scales reflect the different frequencies in the datasets.

**Table 1 cancers-14-03820-t001:** Role of DDX3X in cancer.

Cancer	Role ^1^	Mechanism	Ref.
Breast	+	downregulates KLF4 expression; modulates E-cadherin via HIF-1α; activates ERα	[20,21,22]
Colorectal	dual	+: upregulated KRAS leads to HIF-1α/Yap1 or β-catenin/ZEB1 pathway; −: DDX3X/Snail/E-cadherin axis	[23,24,25]
Lung	dual	+: activates Wnt/β-catenin pathway −: enhances p53-activated p21 transcription; prevents E-cadherin degradation by MDM2 transcription	[26,27,28,29]
Hepatocellular carcinoma	−	overexpressed in HCC cells but blocks cell cycle progression via regulation of cyclin D1 and p21; expression of miRNA; Rotterlin upregulation of DDX3X	[30,31,32,33,34]
Prostate cancer and Ewing sarcoma	+	inhibition of DDX3X leads to a decrease in cellular proliferation	[35,36,37]
Oral squamous cell carcinoma	dual	+: high DDX3X expression is associated with cancer progression in smokers; promotes AREG translation −: low DDX3X expression is associated with poor prognosis in non-smoker patients	[38,39,40]
Head and neck squamous cell carcinoma	+	forms the CBC/DDX3X/eIF3 complex to promotes ATF4 translation	[41]
Glioblastoma	+	high DDX3X level is linked with poor prognosis	[42,43]
Medulloblastoma	+	mutated DDX3X activates WNT/β-catenin signaling and drives stress granules formation	[44,45,46]
Gallbladder carcinoma and	+	high DDX3X level is linked with poor prognosis	[47]
Pancreatic ductal adenocarcinoma	+	high DDX3X level is linked with poor prognosis; promotes p62 accumulation	[48,49]
Chronic myeloid leukemia	+	mutated DDX3X is associated with poor prognosis	[49]
Melanoma	−	promotes MITF translation	[50,51]
NK/T-cell lymphoma	−	DDX3X mutant alters NF-κB and MAPK pathways and increases STAT3/p42/p44 phosphorylation	[52,53]

^1^ Oncogene (+), oncosuppressor (−).

**Table 2 cancers-14-03820-t002:** Role of DDX5 in cancer.

Cancer	Role ^1^	Mechanism	Ref.
Colorectal	+	DDX5 overexpression promotes cancer by AKT/mTOR signaling or association with AldoA	[62,63,64,65,66]
Breast	+	upregulates a subset of miRNAs; highly correlated with Ki67; involved in β-catenin/Wnt pathway	[67,68,69]
Leukemia	+	DDX5 depletion selectively induces stress in AML cells; positive regulator of NOTCH1 signaling; DDX5 inhibition reduces tumor proliferation	[70,71,72]
Non-small-cell lung/small cell lung	+	induces β-catenin to promote cell proliferation	[73,74]
Osteosarcoma	+	lncRNA DLEU1/miR-671-5p/DDX5 interaction to promote cancer progression	[75]
Prostate	+	lncRNA CCAT1/DDX5/miR-28-5p interaction to promote cancer progression; DDX5-ETV4 fusion protein	[76,77]
Gastric	+	high DDX5 expression activates mTOR/SK61 pathway to induce cancer progression; lnc MIAT interaction	[78,79]
Glioblastoma	+	NF-κB p50 subunit activation; lncRNA LINC01116 interaction; hyperactivation of ERK and downregulation of DUSP5	[80,81,82]
Cervical	+	stimulation of the expression of TGF-β1 in CaSki cells	[83]
Endometrial	+	HDGF/DDX5 interaction to induce β-catenin	[84]
Squamous cell carcinoma (HNSCC)	+	high DDX5 expression is associated with cancer progression	[85]
Squamous cell carcinoma (ESCC)	+	decreased DDX5 expression is associated with inhibition of cancer progression	[86]
Human hepatocellular carcinoma (HCC) associated with HBV	−	inhibits cancer progression by interacting with p62/SQSTM1, by regulation of miRNAs and by associating with lncRNA HOTAIR	[87,88,89,90,91]

^1^ Oncogene (+), oncosuppressor (−).

**Table 3 cancers-14-03820-t003:** Selected common interactors of DDX3X and DDX5 and their roles in tumorigenesis.

Interactor	Function	Tumor-Specific Interactome	Role ^1^	Mechanism	Roles of DDX3X/DDX5 in the Same Tumor ^1^	Ref.
RPA2	ss DNA-binding protein	Breast Cancer	+	NF-κB activation	+/+	[92,93]
CTCF	Transcriptional coactivator	Breast Cancer	−	Regulation of transcription	+/+	[94,95]
DHX9	RNA/DNA helicase	Breast Cancer	+	Upregulation of lncRNA	+/+	[96,97]
CUL1	Ubiquitination cofactor	Breast Cancer	+	Positive regulation of proliferation	+/+	[98]
CUL5	Ubiquitination cofactor	Breast Cancer	−	Negative regulation of proliferation	+/+	[99]
YBX1	Transcription factor	Breast Cancer	+	Interaction with lncRNA AC073352.1	+/+	[100]
RPS6KB2	Protein kinase	Breast Cancer	+	Activation of estrogen receptor-alpha	+/+	[101,102]
AGR2	Disulfide isomerase	Breast Cancer	+	Aberrant protein maturation in the ER	+/+	[103,104]
YY1	Transcription factor	Cervical cancer	Dual	Gene-specific recruitment of transcriptional activators or repressors	?/+	[105]
CHD4	Chromatin remodeling	Endometrial cancer	−	Activation of the TGF-β pathway	?/+	[106]
TAF15	mRNA metabolism	Gastric cancer	+	Modulation of stress response	−/+	[107]
MATR3	DNA/RNA-binding protein	Lung cancer	+	Undetermined	Dual/+	[108,109,110]
DYRK2	Protein kinase	Glioblastoma	−	Negative regulation of cell migration	+/+	[111]
PINK1	Mitochondrial protein kinase	Glioblastoma	−	Negative regulation of oxidative stress	+/+	[112]
SYNCRIP	RNA-binding protein	Leukemia	+	Activation of oncogenes	+/+	[113]
ITGA4	Integrin	Leukemia	+	Upregulation of BCL-2/negative regulation of apoptosis	+/+	[114]
MED12	Protein kinase	Prostate cancer	+	Activation of Wnt/β-catenin and TGF-β signaling	+/+	[115]
CNEPA	Kinetochore subunit	Liver cancer	+	Aberrant chromosomal segregation	?/−	[116]

^1^ (+) oncogene, (−) oncosuppressor; (dual) oncogene or oncosuppressor in different studies; (?), role unknown.

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
