# Peer review of "DEAD-Box RNA Helicases DDX3X and DDX5 as Oncogenes or Oncosuppressors: A Network Perspective"

_cancers, 2022, doi:10.3390/cancers14153820_

Round 1

Reviewer 1 Report

Strengths.

·       This review article by Seccchi et al. is timely and important. The authors comprehensively survey the DEAD-box helicase literature cataloguing the roles of these protein as oncogenes and/or oncosuppressors.  The authors chose DDX3X and DDX5 as representative examples. The apparently opposite roles that each helicase can play in tumorigenesis depending on cell/tissue context is an emerging complexity observed among DEAD-box helicases, as well as related DEAH-box helicases. To my knowledge, this is the first review to focus on the dichotomous effects these enzymes can have on tumors, and as such, will be important for designing drugs that target these helicases. Given the rising interest in DEAD/DEAH-box helicases as targets of potential cancer therapies, this review provides critical insights for the rational design of therapeutic strategies.

·       Additionally, the authors provide an interesting network analysis and comparison of DDX3X and DDX5 using publicly available databases, adding novel insights/data into this review article. This approach also provides a helpful model for others to follow to perform additional network analysis for other DEAD/DEAH-box helicases.

·       Finally, the authors effectively highlight how important the interactome is in mediating the different functions of DDX3X and DDX5. This is a key piece of insight for understanding the cell/tissue context dependent functions.

Critiques/suggestions.

·       Its is not clear why DDX3X and DDX5 were chosen as the representative examples or why these two were directly compared. Why not DEAD- and not DEAH-box helicases? Some brief language in the Introduction should be added.

·       It would be helpful to the reader, especially when reading the introduction, to provide protein domain diagrams for DDX3X and DDX5, highlighting the major domains. It would be interesting to also include in this diagram the location/residues that are commonly mutated in cancer.

·       Differences in y-axis scales  in Figure 1 are potentially misleading.

·       Section 4:

o   Overall, this section needs a more detailed description of how the analysis was performed. For example, what types of cancer are the interactome data taken from? If the interactomes are merged from multiple data sets and tumor types, then include that in the description and include how they were pooled.

o   Also, some brief information about how the two databases construct the interactomes would be helpful to the readers, especially those that might want to repeat this type of analysis with other DEAD/DEAH-box helicases.

·       Overall, there are numerous long summaries of the relevant literature with short summaries of discussion points at the end of each sub section. This is effective; however, I would have liked to have a seen a more developed “big picture discussion” in the Conclusion section. For example,  “several working hypotheses” is mentioned in the Conclusion but the hypotheses are not restated, elaborated on, or an attempt to merge them into a working model is not included.

·       Table 3.

o   What was the criteria for selecting the interactors to include in this table? Was a non-biased approach utilized (e.g. the top 20 statistical hits)? If not, then the table of interactors could potentially be misleading.

Editorial edits

·       Overall the writing is good, but could use some careful copy editing before publication. Below are some examples, but is likely not a comprehensive list of typo’s.

·       Be consistent with either DDX3X or DDX3 throughout

·       Simple Summary section:

o   Period after “tumor” in second to last sentence

·       Introduction:

o   last sentence of first paragraph, “Lately,” should be “Recently,”

o   last sentence of introduction is a run-on

·       Section 2.4

o   First sentence in abrupt/choppy

·       Section 2.7

o   “proliferations and progressions” should not be plural? (double check)

·       Section 4.1

o   “specular” should be “special”

·       Conclusion

o   first sentence is a run-on

·       There is an extra space in between oncogenic and lncRNA on page 9 in the top paragraph.

·       On page 10 and 11 the websites are in much bigger font than everything else.

Author Response

Answers to Reviewers

Manuscript ID: cancers-1837638 R1

Reviewer 1

Critiques/suggestions.

Q1. It is not clear why DDX3X and DDX5 were chosen as the representative examples or why these two were directly compared. Why not DEAD- and not DEAH-box helicases? Some brief language in the Introduction should be added.

A1. We thank the Reviewer for this very appropriate remark. We have selected DDX3X and DDX5 since they satisfied the two main criteria we have set for our analysis: a robust and extended literature describing their roles in different cancers and comprehensive proteomics data. Indeed, these two proteins are so far the best characterized members among the DEAD-box RNA helicase family. We have clarified the reasons of our choice in the introduction of the revised version.

Q2. It would be helpful to the reader, especially when reading the introduction, to provide protein domain diagrams for DDX3X and DDX5, highlighting the major domains. It would be interesting to also include in this diagram the location/residues that are commonly mutated in cancer.

A2. We thank the reviewer for this suggestion. We have generated a new Figure 1 where we show the available crystal structures of DDX5 (N-terminal domain, Fig.1A) and DDX3X (helicase core, Fig. 1B), in order for the reader to appreciate the conserved RecA N-ter domain folding of both proteins, and a schematic representation of the conserved helicase motifs with the respective functional roles (Fig. 1C). Unfortunately, the mutations occurring in these proteins have not been characterized for all cancer types. In addition, in most cases the alterations observed are at the expression levels. So, in order not to introduce further complexity in the figure, which would have been in any case only partially informative, we have elected not to include the mutations.

Q3. Differences in y-axis scales in Figure 1 are potentially misleading.

A3. We thank the Reviewer for this observation. We have included in the figure legend (now Figure 2 of the revised manuscript) an explanatory note so that the reader will not be misled by the difference.

Q.4 Section 4:

o   Overall, this section needs a more detailed description of how the analysis was performed. For example, what types of cancer are the interactome data taken from? If the interactomes are merged from multiple data sets and tumor types, then include that in the description and include how they were pooled.

o   Also, some brief information about how the two databases construct the interactomes would be helpful to the readers, especially those that might want to repeat this type of analysis with other DEAD/DEAH-box helicases.

A4. We agree with the Reviewer that this point is important and we thank the Reviewer for bringing it up. We have added in the revised version a more detailed description of the rationale used for the construction of the interactome lists. We would like to remark that both databases are public repositories that collect and made available data on protein interactions from model organisms. Deposited data provide manually curated datasets of protein interactions derived from high-throughput analyses and from the literature; protein interaction data refer to a wide range of cellular and tissue systems and are not restricted to any pathological condition. We have then conducted a literature search for the identified proteins in order to find those for which robust literature data were available supporting their roles in different cancer types.

Q5. Overall, there are numerous long summaries of the relevant literature with short summaries of discussion points at the end of each sub section. This is effective; however, I would have liked to have a seen a more developed “big picture discussion” in the Conclusion section. For example,  “several working hypotheses” is mentioned in the Conclusion but the hypotheses are not restated, elaborated on, or an attempt to merge them into a working model is not included.A5. We thank the Reviewer for this important observation. In the revised manuscript we added in the conclusion a new section focusing on the interactions of both DDX5 and DDX3X with shared components of two main proliferative signaling pathways, often deregulated in cancer: AKT/mTOR and Wnt/b-catenin, as revealed by our analysis, highlighting the potential regulatory roles of the two helicases. In addition, we brought to the readers’ attention the contribution of DDX3X and DDX5 in miRNA biogenesis, particularly on subsets of cancer-specific miRNAs. We have added additional relevant references to support our hypotheses.

Q6. Table 3.

o   What was the criteria for selecting the interactors to include in this table? Was a non-biased approach utilized (e.g. the top 20 statistical hits)? If not, then the table of interactors could potentially be misleading.

A6. We thank the Reviewer for this remark. We have selected those interactors present in the two databases of interest, for which robust literature data were available to support their roles in cancer progression, giving us the opportunity to explore possible links with DDX3X and DDX5 based on solid data. We have clarified this in the revised manuscript.

Q7. Editorial editsA7. We thank the Reviewer for the careful reading and we apologize for the imprecisions. We have inserted/corrected all the requested modifications in the revised text. As for the first sentence of the Conclusion section, we reasoned that it would have been better suited for the Introduction and we have thus moved it up in a shortened form in our revised text.

Reviewer 2 Report

Dear authors,

I read your review on the roles of the DEAD box RNA helicases DDX3X and DDX5 in cancer with great interest. The review is well presented, topical and informative. My only suggestion is to consider including a structural image of the conserved DEAD box helicase structure as Figure 1 at the beginning of the review (for example https://pubmed.ncbi.nlm.nih.gov/20941364/). This would make the enzymes more tangible for the reader.

Author Response

Answers to Reviewers

Manuscript ID: cancers-1837638 R1

Reviewer 2

Q1. My only suggestion is to consider including a structural image of the conserved DEAD box helicase structure as Figure 1 at the beginning of the review.

A2. We thank the reviewer for this suggestion. As requested, we have generated a new Figure 1 where we show the available crystal structures of DDX5 (N-terminal domain, Fig.1A) and DDX3X (helicase core, Fig. 1B), in order for the reader to appreciate the conserved RecA N-ter domain folding of both proteins, and a schematic representation of the conserved helicase motifs with the respective functional roles (Fig. 1C).

Reviewer 3 Report

The Review “DEAD box RNA helicases as oncogenes or oncosuppressors: a network perspective” by Secchi et al. is a comprehensive summary of the current literature on DDX3X and DDX5 and their function in several cancers. Furthermore, the authors define protein interaction partners of DDX3X and DDX5 and suggest new cancer specific hypothesis by these proteins. 

11.)   The authors discuss the function of DEAD box RNA helicases DDX3X and DDX5 in cancer in their presented review here. In this light, the title of the manuscript is misleading. Please specify the title as the review summarizes current knowledge on DDX3X and DDX5 and not other DEAD box RNA helicases. 

22.) Please mention in the introduction the function of DEAD box RNA helicases in innate immunity and viral infections.

33.) Please change “DDX3” (“…have demonstrated the roles of DDX3 in cancer biology (15)., “High DDX3expression is related with poor survival in smoker patients (35). ”(39). In these cancer types, high DDX3expression level…”) to DDX3X. 

44.)  In chapter 2.6 not all abbreviation are explained, like OSCC in the heading and CBC later in the text. Please change accordingly. 

55.) Reviews should provide a comprehensive foundation of a topic, explain current state if knowledge, identify gaps in existing studies and highlight new technologies or methodologies. Therefore, the unique interactors of DDX3X/DDX5 and their potential biological relevance in different cancer types (starting in chapter 4.2) should be shortened as this is no knowledge and statements by the authors are only hypothesis based. 

Author Response

Answers to Reviewers

Manuscript ID: cancers-1837638 R1

Reviewer 3

Q1. The authors discuss the function of DEAD box RNA helicases DDX3X and DDX5 in cancer in their presented review here. In this light, the title of the manuscript is misleading. Please specify the title as the review summarizes current knowledge on DDX3X and DDX5 and not other DEAD box RNA helicases.

A1. We thank the Reviewer for this very appropriate comment. Accordingly, we have changed the title into “DEAD box RNA helicases DDX3X and DDX5 as oncogenes or oncosuppressors: a network perspective” so as to avoid any misunderstanding.

Q2. Please mention in the introduction the function of DEAD box RNA helicases in innate immunity and viral infections.

A2. We thank the Reviewer for this suggestion. In the Introduction we now say “RNA helicases are involved in many biological processes, such as cell cycle regulation, stress response, apoptosis, innate immunity and virus infection”. Since, however, those latter aspects were not the focus of the Review, we did not provide further details, but we have changed the corresponding reference (Ref.6) with one providing a comprehensive overview of their roles in immunity and infection.

Q3. Please change “DDX3” (“…have demonstrated the roles of DDX3 in cancer biology (15)., “High DDX3expression is related with poor survival in smoker patients (35). ”(39). In these cancer types, high DDX3expression level…”) to DDX3X.

Q4. In chapter 2.6 not all abbreviation are explained, like OSCC in the heading and CBC later in the text. Please change accordingly.

A3-4. We thank the Reviewer for the careful reading and we apologize for these oversights. We have now changed to DDX3X everywhere in the manuscript and defined the acronyms where needed as requested.

Q5. Reviews should provide a comprehensive foundation of a topic, explain current state if knowledge, identify gaps in existing studies and highlight new technologies or methodologies. Therefore, the unique interactors of DDX3X/DDX5 and their potential biological relevance in different cancer types (starting in chapter 4.2) should be shortened as this is no knowledge and statements by the authors are only hypothesis based.

A5. We thank the Reviewer for this comment. We absolutely agree that one of the aims of a review article is to present the current state-of-the-art and its eventual limitations. However, we believe that another scope of a review article is also the elaborate on the current knowledge in order to highlight connections between different research works in the same field, so as to put forward novel hypotheses and suggestions for future work. That was one of the aims of our manuscript, to broaden the perspective and suggest new venues of research. So, in order not to be misleading towards the readers, we have now added short introductory sentences to Sections 4.2 and 4.3, to clearly state that we were proposing working hypotheses, based on the current knowledge, and not established facts.